# Prevalence of latent tuberculosis infection among health workers in Afghanistan: A cross-sectional study

Ghulam Qader Qader[1]*, Mohammad Khaled Seddiq[2], Khakerah Mohammad Rashidi[1], Lutfullah Manzoor[2], Azizullah Hamim[1], Mir Habibullah Akhgar[2], Laiqullrahman Rahman[2], Sean Dryer[3], Mariah Boyd-Boffa[4], Aleefia Somji[3], Muluken Melese[3], Pedro Guillermo Suarez[3]

1 Challenge TB Project, Management Sciences for Health, Kabul, Afghanistan, 2 National Tuberculosis Program, Ministry of Public Health, Kabul, Afghanistan, 3 Management Sciences for Health, Arlington, VA, United States of America, 4 Management Sciences for Health, Medford, MA, United States of America

* gqader@gmail.com

## Abstract

### Background

About 26% of the world's population may have latent tuberculosis infection (LTBI). Health care workers are a high-risk category because of their professional exposure.

### Methods

This cross-sectional study assessed the LTBI burden among health care workers in Afghanistan, a high-TB-burden country. We selected health facilities using a systematic sampling technique and invited all workers at the targeted health facilities to participate. Participants were interviewed about sociodemographic and exposure variables and received tuberculin skin tests for LTBI.

### Results

Of the 4,648 health care workers invited to participate, 3,686 had tuberculin skin tests. The prevalence of LTBI was found to be 47.2% (1,738 workers). Multivariate analysis showed that a body mass index of ≥ 30 and marriage were associated with an increased risk of LTBI. Underweight (body mass index of ≤ 18 and below) and normal body mass index had no association with increased risk of LTBI.

### Conclusion

LTBI is high among health care workers in Afghanistan. We recommend instituting infection control measures in health facilities and screening workers for timely TB diagnosis.

**Data Availability Statement:** All relevant data are within the paper.

**Funding:** The United States Agency for International Development (USAID) funded this

study through the Challenge TB project under cooperative agreement number AID-OAA-A-14-00029, and the Global Fund to Fight AIDS, Tuberculosis and Malaria provided funding for this study. The contents of the article are the responsibility of the authors alone and do not necessarily reflect the views of USAID or the US government, or the Global Fund . The publication fee is covered by the USAID funded Sustainable Technical and Analytic Resources (STAR) project, through Public Health Institute (PHI).

**Competing interests:** The authors have declared that no competing interests exist.

## Background

In 2017, 10 million people developed tuberculosis (TB) and 1.3 million succumbed to the disease [1]. An estimated 2 billion people worldwide—26% of the global population—are believed to be infected with TB [1]. Of the 10 million estimated TB incident cases, about 4.4 million prevalent cases are found in the South-East Asia region [1]. Afghanistan has a prevalence rate of 340 and incidence rate of 189 per 100,000 population. In 2019, Afghanistan's National TB Program (NTP) diagnosed and reported 48,420 TB cases of all forms (73% of estimated incident cases) [1]. Although Afghanistan has a treatment success rate of 91%, the World Health Organization (WHO) estimates that 12,000 Afghans still die of TB annually, although in 2019 TB mortality declined to 9,800 [1, 2].

The only sign of TB infection is a positive reaction to the tuberculin skin test (TST) or a blood test. Persons with latent TB infection (LTBI) are not infectious, but having a TB infection is a precondition for having TB disease, and 5% to 10% of those infected could develop the disease in their lifetimes. The majority of those cases will develop within five years of infection [3]. Twenty-three percent of the world's population is estimated to have LTBI (based on TST), which represents a huge reservoir of potential TB disease and is therefore a challenge to TB control [4].

To control TB effectively, it is important to know the burden of TB infection among health care workers (HCWs), who are at higher risk of TB infection due to exposure to diagnosed and undiagnosed TB patients [5–8]. This risk is proportionally more alarming in low- and middle-income countries because of both increased exposure and lack of preventive measures, such as poor workplace ventilation and inadequate precautions during sputum collection and bronchoscopy [9]. Occupational risk factors associated with LTBI include duration of employment in a health care profession [5, 10–14]; being a nurse, diabetic, or smoker [11]; being over the age of 35 [15, 16]; employment in cleaning or housekeeping and in a health care setting with high patient turnover [17]; having been employed in an HIV clinic or ward [12]; and not having had a Bacille Calmette-Guérin vaccination or being immunocompromised [18].

In a systematic review of TB studies among HCWs in South Africa, most reflected a higher incidence and prevalence of active TB disease in HCWs, including drug-resistant TB, compared to the surrounding community or general population [19]. A review of HCWs in seven countries with high TB prevalence (> 100/100,000 population) reported an LTBI prevalence rate of 47% [20].

Through this study, we sought to address the lack of information on estimates of TB infection among HCWs in Afghanistan. The findings can assist the NTP and Ministry of Public Health to shape policy to ensure a safer working environment for HCWs.

## Methods

### Study design and population

This cross-sectional study among HCWs was conducted in 23 provinces of Afghanistan between September and December 2017. "Health care workers" for this study included any worker in the health facility: doctors, nurses, laboratory professionals, midwives, vaccinators, community health workers, cleaners, guards, administrative staff, and others. The sample size for the study was determined using a single proportion formula [21] assuming an LTBI prevalence (pone') of 47% among HCWs [20], type I error of 5%, and desired precision of 0.05, and a response rate of 60%. The resulting sample size was 1,281 HCWs. All 2,499 health facilities in Afghanistan—public, private, and prison—are listed in the national health information system register and a systematic random sampling technique to select study facilities. Assuming an

average number of five HCWs per health facilities, and no difference in LTBI between health facilities, the total number of health facilities required to obtain the sample of HCWs was 249. Health facilities were listed in alphabetical order and every 10th health facility was included in the study, the first one being picked by lottery method among those health facilities listed from 1–10. All HCWs in the randomized health facilities were then included as potential participants.

## Training of data collectors

The data collection tool was developed in English first and translated into local languages (Dari and Pashto) and then retranslated into English by a third person who was not involved in the drafting of the first version to check the consistency of the questionnaires. The local language version was then administered to 50 health workers to check for clarity, consistency, and ease of understanding. Following this the actual data collectors, who were unemployed HCWs, were trained to pilot-test the questionnaires, which included sociodemographic and exposure variables, with study participants. Physicians with experience in administering TST were trained by the principal investigator (GQ) to follow study procedures under supervision to administer the test to 120 HCWs, as indicated in the study procedures below. The inter-observer agreement level for TST skin induration measurement was determined and the weighted Kappa value was 0.75. Twenty-two physicians with perfect skills were selected to conduct TSTs for the study.

## Study procedures

Data collectors administered the questionnaires, which included HCWs' history of active TB and of TSTs, followed by screenings for active TB symptoms and signs, such as a cough lasting two weeks or more, night sweating, and weight loss. Those with signs of TB were asked to give a sputum sample for GeneXpert testing (Xpert MTB/RIF assay, Cepheid, Sunnyvale, USA). All HCWs with TB constitutional symptoms also received a chest X-ray.

HCWs without signs or symptoms of TB and with no history of active TB were administered 0.1 ml of 5 tuberculin units of purified protein derivative (BB-NCIPD Ltd., 1504 Sofia, Bulgaria) intradermally in the volar surface of the forearm [22]. Trained physicians read the TST induration within 48–72 hours of administration.

## Exclusion criteria

HCWs with presumptive TB and those with a history of active TB were excluded, as were those who did not volunteer for the TST test.

## Interpretation of TST results

Per the NTP's definition, the following cutoff points were used to interpret the skin test: CHWs with induration of $\geq$ 10 mm at 48–72 hours were considered to have LTBI, those with a reading of 0–9 mm and were HIV negative were considered negative for LTBI. If the CHW was HIV positive, 5–10 mm was also considered LTBI [22].

## Data entry and analysis

The data were entered in Epi Info version 7.2.2.6 and transferred to SPSS version 23 for analysis. A univariate analysis was used to identify potential variables predicating positive TST with Pearson's chi-square test, and all variables with a P value of less than 5% in the bivariate analysis were further analyzed with multiple logistic regression analysis to determine any

associations. Adjusted odds ratios (AORs) and 95% confidence intervals (CIs) were used to determine the association with sociodemographic and exposure variables. In the calculation of the odds ratios, missed variables were excluded.

### Ethical considerations

The study was approved by the Afghan Ministry of Public Health Institutional Review Board under approval number 43864, dated August 2, 2017. Study participants were properly instructed in local languages (Dari or Pashto) about the study purpose, the procedures to administer TST and possible side effects related to the injection site. After these explanations, all the study participants gave written consent in Dari or Pashto. Their information was kept confidential and not shared with anyone outside of the study team. The data set had no personal identifiers. Those who did not consent were excluded from the study. Those with signs of TB were provided a free diagnosis, and those diagnosed with TB were given free treatment, according to national guidelines.

## Results

In total, the 249 health facilities sampled had 4,648 health workers, and, of these, 3,975 (85.5%) consented to participate; 154 (3.3%) refused to participate; 474 (10.1%) were excluded from the study because they were presumptive TB cases; and another 45 (0.9%) had a history of TB. We gave TSTs to 3,975 HCWs and read results for 3,686 (92.7%); 289 HCWs (7.3%) did not return for the reading and were excluded from the analysis (Fig 1).

In this study, male HCWs constituted 2,573 (69.8%) of participants. The mean age of study participants was 34.5, with a median age of 32. Of the participants, 2,923 (79.3%) were married, 2,152 (58.4%) were college or university graduates, and the average monthly income of 1,616 (43.8%) was less than US$130. In 1,966 (53.3%) of participants, body mass index (BMI) was normal (18–24.9); 1,195 (32.4%) were overweight (BMI = 25.0–29.9); 237 (7.4%) were obese (BMI = $\geq$ 30); 115 (3.1%) were underweight (BMI = < 18); and in 134 HCWs (3.6%), measurements were not taken.

Of 3,686 test results, 1,738 were LTBI positive, for a prevalence of 47.1% (95% CI 45.4%-48.7%). Men had a higher TST positivity—a total of 1,237 (71.3%)—than women, of whom only 497 (28.5%) tested positive, but the difference was not statistically significant (P > 0.05). In the bivariate analysis, the only groups that showed significant associations with LTBI were married (OR = 2.13, 95% CI 1.77–2.57), aged between 35 and 44 (OR = 1.30, 95% CI 1.07–1.57), and illiterate (OR = 1.34, 95% CI 1.06–1.7) (Tables 1 and 2).

In the exposure variables, those who had worked in health care for less than 10 years had a higher prevalence of LTBI—1,115 (64.1%)—and the difference was significant (P = 0.04). The prevalence of LTBI was not associated with health facility type, cigarette smoking, or family history of TB (P > 0.05). There were only four reported cases of HIV-positive health workers; the majority (56.7%) of HCWs did not know their HIV status (Table 3).

In the multiple logistic regression analysis, only BMI values of 30+ (AOR 1.32, 95% CI 1.01–1.73) and being married (AOR 1.99, 95% CI 1.65–2.4) were associated with LTBI (Table 2). Years of service was not associated with LTBI (AOR 1.18, 95% CI 0.97–1.44)].

## Discussion

The prevalence of LTBI among health workers in Afghanistan in this study was 47.1%, which is more than double the 23% reported among the global population [23]. There are no recent studies for Afghanistan, but two studies in 1958 and 1978 among children showed that the annual risk of infection was 3.5% [24]. In another survey among six- and seven-year-old

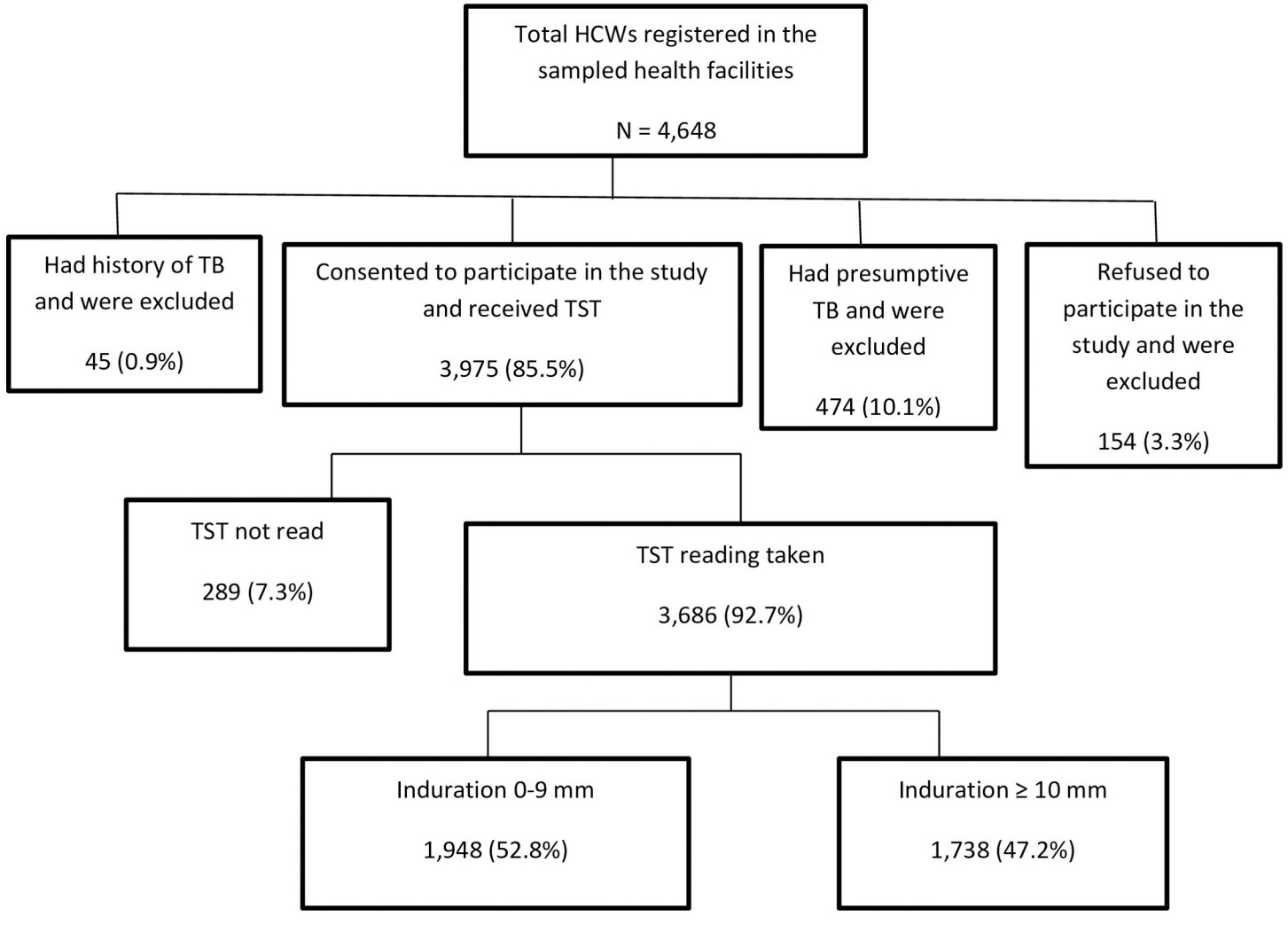

**Fig 1. Flow chart showing health care workers who participated in the LTBI study in Afghanistan.**

children in Kabul, the prevalence of TB infection at a cutoff point of TST 8 mm and above was 4.3% [25].

Our findings are similar to those of studies of HCWs in other high-TB-burden countries. A systematic review of seven countries reported LTBI prevalence in HCWs ranging from 37% in Brazil to 64% in South Africa [21]. LTBI prevalence among HCWs was 36.8% in an Indian tertiary hospital [26]; in Côte d'Ivoire it was 79% [27]; 58.8% in Turkey [28]; 57% in Uganda [29]; and 39.4% in Brazil [30]. In neighboring Iran, however, the prevalence was 24.8% for high-risk workers such as laboratory professionals and 14.8% for low-risk HCWs [31]. Iran and Turkey, however, are low-TB-burden countries, with an incidence of 14 and 17/100,000 respectively [1].

In our study, administrative staff had a higher rate of LTBI, at 35%, followed by 19.3% for nurses, but the difference was not statistically significant (P = 0.2). Other studies reported that housekeeping staff, older health workers, and those working in areas of high patient turnover had higher rates of LTBI [6, 17, 32, 33]. The high LTBI in cleaners, guards, and other administrative staff is probably due to their exposure to patients at registration and in waiting areas as well as high exposure to sputum during cleaning and garbage disposal.

**Table 1. Sociodemographic variables and association with LTBI among health care workers in Afghanistan (N = 3,686).**

| Variables | | TST Positive (≥ 10 mm) | TST Negative (0–9 mm) | P Value |
|---|---|---|---|---|
| Sex | Male | 1,237 (71.1%) | 1,336 (68.8%) | P = 0.054 |
| | Female | 497 (28.6%) | 601 (30.9%) | |
| | Missing data | 4 (0.2%) | 11 (0.6%) | |
| | Total | 1,738 (47.1%) | 1,948 (52.9%) | |
| Age | 18–24 | 372 (21.4%) | 441 (22.6%) | P = 0.025 |
| | 25–34 | 530 (30.5%) | 601 (30.9%) | |
| | 35–44 | 435 (25.1%) | 398 (20.4%) | |
| | 45–54 | 260 (15.0%) | 336 (17.2%) | |
| | 55–64 | 109 (6.3%) | 128 (6.6%) | |
| | ≥ 65 | 14 (0.8%) | 20 (1.0%) | |
| | Missing data | 15 (0.9%) | 39 (1.2%) | |
| | Total | 1,738 (47.1%) | 1,948 (52.9%) | |
| Marital status | Married | 1,490 (85.9%) | 1,433 (73.6%) | P < 0.001 |
| | Single | 208 (12.0%) | 481 (24.7%) | |
| | Separated | 1 (0.1%) | 3 (0.2%) | |
| | Not mentioned | 27 (1.6%) | 15 (0.8%) | |
| | Missing data | 9 (0.5%) | 16 (0.8%) | |
| | Total | 1,735 (47.1%) | 1,948 (52.9%) | |
| Highest level of education completed | Illiterate | 192 (11.1%) | 170 (8.7%) | P = 0.005 |
| | Primary school | 198 (11.4%) | 199 (10.2%) | |
| | Secondary school | 259 (14.9%) | 289 (14.8%) | |
| | College or university graduate | 962 (55.4%) | 1,190 (61.1%) | |
| | Other | 12 (0.7%) | 5 (0.3%) | |
| | Missing data | 115 (6.6%) | 95 (4.9%) | |
| | Total | 1,738 (47.1%) | 1,948 (52.9%) | |
| Professional category | Doctor | 224 (12.9%) | 237 (12.2%) | P = 0.205 |
| | Nurse | 336 (19.3%) | 334 (17.1%) | |
| | Midwife | 151 (8.7%) | 219 (11.2%) | |
| | Laboratory professional | 95 (5.5%) | 101 (5.2%) | |
| | Community health worker | 67 (3.9%) | 68 (3.5%) | |
| | Pharmacy professional | 46 (2.7%) | 55 (2.8%) | |
| | Vaccinator | 123 (7.1%) | 114 (5.9%) | |
| | Admin. and support staff | 607 (35.0%) | 715 (36.7%) | |
| | Other | 70 (4.0%) | 83 (4.3%) | |
| | Missing data | 19 (1.0%) | 22 (1.1%) | |
| | Total | 1,738 (47.1%) | 1,948 (52.9%) | |
| Monthly income (in Afghani) | < 10,000 | 777 (44.8%) | 839 (43.1%) | P = 0.4 |
| | 10,000–20,000 | 664 (38.3%) | 754 (38.7%) | |
| | 20001–30,000 | 161 (9.3%) | 168 (8.6%) | |
| | 30001–50,000 | 43 (2.5%) | 50 (2.6%) | |
| | > 50,000 | 18 (1.0%) | 10 (0.5%) | |
| | Did not specify | 57 (3.3%) | 75 (3.9%) | |
| | Missing data | 15 (0.9% | 52 (2.7%) | |
| | Total | 1,735 (47.1%) | 1,948 (52.9%) | |

(*Continued*)

**Table 1.** (Continued)

| Variables | | TST Positive (≥ 10 mm) | TST Negative (0–9 mm) | P Value |
|---|---|---|---|---|
| BMI (kg/m2) | < 18 | 43 (2.5%) | 72 (3.7%) | P < 0.001 |
| | 18–24 | 882 (50.8%) | 1,084 (55.6%) | |
| | 25–29 | 603 (34.8%) | 592 (30.4%) | |
| | 30+ | 151 (8.7%) | 122 (6.3%) | |
| | Missing data | 56 (3.2%) | 78 (4.0%) | |
| | Total | 1,735 | 1,948 | |
| History of TST | Yes | 15 (0.9%) | 15 (0.8%) | P = 0.448 |
| | No | 1,593 (91.6%) | 1,787 (91.7%) | |
| | I do not know | 130 (7.4%) | 182 (9.3%) | |
| | Total | 1,738 | 1,948 | |

In the bivariate analysis, the factors associated with TB were age, years of professional experience, BMI, marital status, and education level. But in the multivariate analysis, the variables associated with LTBI were BMI and marital status. The AOR for obese participants (BMI ≥ 30) was 1.32 (95% CI 1.01–1.73), while the AOR for those with BMI between 18.0 and 24.9 was 0.83 (95% CI 0.72–0.97). There was no association in our study between LTBI and underweight, a finding which is in line with those of other studies; however, unlike LTBI, TB disease exhibits high prevalence among malnourished persons [34]. Another article reported that the risk of TB increased by 14% for each point reduction in BMI between 18.0 and 29.9,

**Table 2. Multiple logistic regression analysis of factors associated with LTBI among health care workers in Afghanistan.**

| Variables | | Unadjusted Odds Ratio (95% CI) | Adjusted Odds Ratio (95% CI) |
|---|---|---|---|
| Years of professional service | < 10 years | | 1 |
| | 10–19 years | 1.17 (0.99–1.39) | 1.18 (0.97–1.44) |
| | 20–39 years | 0.97 (0.75–1.25) | 1.04 (0.77–1.39) |
| | 40+ years | | *Insufficient number* |
| BMI (kg/m2) | <18 | | 1 |
| | 18–24 | 1.28 (0.84–1.97) | 1.20 (0.78–1.87) |
| | 25–29 | 1.68 (1.09–2.60) | 1.44 (0.93–2.25) |
| | 30+ | 2.04 (1.26–3.34) | 1.70 (1.04–2.80) |
| Marital status | Single | | 1 |
| | Married | 2.13 (1.77–2.57) | 1.99 (1.65–2.40) |
| | Separated | | *Insufficient number* |
| | Widowed | | *Insufficient number* |
| | Not mentioned | 1.60 (0.77–3.52) | 1.41 (0.66–3.15) |
| Highest level of education completed | Illiterate | | 1 |
| | Primary school | 0.88 (0.64–1.19) | 0.92 (0.67–1.27) |
| | Secondary school | 0.77 (0.58–1.03) | 0.87 (0.65–1.17) |
| | College or university graduate | 0.72 (0.56–0.91) | 0.85 (0.66–1.10) |
| Age | 18–24 | | 1 |
| | 25–34 | 1.05 (0.88–1.26) | 1.02 (0.85–1.23) |
| | 35–44 | 1.30 (1.07–1.57) | 1.18 (0.95–1.46) |
| | 45–54 | 0.93 (0.75–1.15) | 0.88 (0.69–1.12) |
| | 55–64 | 1.07 (0.80–1.44) | 1.02 (0.74–1.40) |
| | ≥ 65 | 0.85 (0.39–1.79) | 0.74 (0.33–1.58) |

**Table 3. Exposure variables and associated factors among health care workers in Afghanistan.**

| Variables | | TST Positive (≥ 10 mm) | TST Negative (0–9 mm) | P Value |
|---|---|---|---|---|
| Years of professional service | < 10 years | 1,115 (64.1%) | 1,293 (66.3%) | P = 0.04 |
| | 10–19 years | 405 (23.3%) | 385 (19.7%) | |
| | 20–39 years | 130 (7.4%) | 169 (8.6%) | |
| | 40+ years | 6 (0.3%) | 10 (0.5%) | |
| | Missing data | 82 (4.7%) | 91 (4.6%) | |
| Type of health facility worked in | Public primary health care units | 628 (36.1%) | 668 (34.3%) | P > 0.05 |
| | Public hospitals | 676 (38.8%) | 820 (42.1%) | |
| | Private health facility | 326 (18.7%) | 341 (17.5%) | |
| | Other health facilities | 43 (2.4%) | 53 (2.7%) | |
| | Not categorized | 65 (3.7%) | 66 (3.4%) | |
| HIV status | Positive | 2 (0.001%) | 2 (0.001%) | P > 0.05 |
| | Negative | 754 (43.3%) | 835 (42.8%) | |
| | Unknown | 982 (56.5%) | 1,111 (57.0%) | |
| Cigarette smoking | Yes | 91 (5.2%) | 114 (5.8%) | P > 0.05 |
| | No | 1,475 (84.8%) | 1,694 (86.9%) | |
| | Other | 172 (9.8%) | 140 (7.1%) | |
| Family TB history | Yes | 78 (4.5%) | 89 (4.6%) | P > .05 |
| | No | 1,624 (93.6%) | 1,818 (93.3%) | |
| | Missing data | 33 (1.9%) | 41 (2.1%) | |

but it not clear whether that is true for those with body mass indexes of < 18 and ≥ 30 [35]. One possible reason for the low LTBI in underweight people could be that impaired immunity in malnourished people prevents reaction to the TST; instead, those with low BMI develop TB disease faster [36]. Cross-sectional studies also cannot show individuals' BMI histories, so creating an association for a chronic infection like LTBI by measuring BMI at a single point in time may not be appropriate and a longitudinal study would be preferable.

The other LTBI association we found is marital status: the AOR was 1.99 (95% CI 1.65–2.40) for married participants compared to 0.47 (95% CI 0.39–0.57) for single HCWs. Although the reason for the high association of LTBI with marriage is unknown, a similar finding has been reported in South Africa [37]. One reason could be linked to family size and therefore household overcrowding, which might facilitate disease transmission if one member of the household develops TB. Further studies are recommended because there is no plausible reason for married couples to have high LTBI.

In high-TB-burden countries like Afghanistan, where LTBI among HCWs is high, this study should spur action to prevent HCWs from developing TB. Although the benefits of preventive therapy are well established [38, 39]—the protection effectiveness of isoniazid (INH) alone was 90% in studies conducted in developing and developed countries [22]—recommendations for treating HCWs in high-TB-burden countries with ongoing transmission are different from those in low-incidence settings. There is no consensus on treating LTBI among HCWs or in communities in high-TB-burden countries [4]. The WHO's recently revised LTBI guidelines conditionally recommend that children, adolescents, and adults of all ages who are household contacts of bacteriologically confirmed pulmonary TB cases receive preventive treatment; however, treatment of HCWs is recommended only for low-TB-incidence countries, in part because the re-infection rate is high without a robust TB control program [3]. The lack of easy LTBI diagnostics is another factor for not treating HCWs in high-TB-burden countries.

There is a single experience of community-wide INH preventive therapy, in Alaska between 1958 and 1964, in which the TST prevalence in participating communities was above 80% and the annual risk of infection was 8% at baseline. In this community, daily INH was given for the intervention and placebo given in the control arm for 12 months. The incidence of the disease in the intervention group fell dramatically, and cumulative TB incidence was reduced by 60% [40].

Despite high re-infection rates in high-TB-burden countries, expanding preventive treatment beyond the current recommendation to include high-risk groups such as HCWs and contacts should be considered. The WHO LTBI guidelines conditionally recommend treating children over the age of five, adolescents, and adults in high-TB-burden countries, and Afghanistan can decide to treat HCWs, as they are a high-risk group [3]. Ample evidence shows that providing INH preventive therapy to HIV/AIDS patients and children under five has reduced TB disease, although re-infection is a possibility [5]. Studies in low-incidence settings have shown that INH prophylaxis reduced the development of active TB by 40% over two years or longer [39]. Another article reported that INH preventive therapy reduces the development of active disease by 60% to 90%, depending on treatment adherence levels [41]. All these data favor providing preventive therapy to high-risk groups like HCWs even in high-TB-burden countries like Afghanistan. Environmental, administrative, and personal protection measures to control TB infection that have been implemented in US health care settings associated with TB transmission among patients and HCWs have significantly reduced infections [42], but similar measures are usually poorly implemented in resource-poor countries.

## Limitations

Our study has some limitations. We did not get complete information for most of the study subjects about Bacille Calmette-Guérin vaccination, which might affect TST results. Another drawback of this study is measuring BMI in a cross-sectional study and creating an association with LTBI. Cross-sectional studies cannot show individuals' BMI histories, so creating an association for a chronic infection such as LTBI by measuring the BMI at a single point in time may not be appropriate. We did not include other factors that affect LTBI, such as diabetes mellitus, malignancies, HIV, and other diseases, in our study.

## Conclusion

LTBI in HCWs in Afghanistan is very high. Until evidence is generated globally on the effectiveness of targeted treatment of LTBI in high-TB-burden countries or on new sensitive and specific LTBI diagnostics or shorter treatments, we recommend regular screenings of HCWs for active TB and early treatment. Another recommendation is to strengthen the environmental, administrative, and personal protection measures against TB infection that are recommended by the WHO [43].

## Acknowledgments

We thank the Afghanistan Ministry of Public Health and NTP and all the health workers, laboratory professionals, and TB program coordinators who participated in this study. Barbara K. Timmons edited the manuscript.

## Author Contributions

**Conceptualization:** Mohammad Khaled Seddiq, Khakerah Mohammad Rashidi, Azizullah Hamim, Mir Habibullah Akhgar, Laiqullrahman Rahman, Muluken Melese, Pedro Guillermo Suarez.

**Formal analysis:** Ghulam Qader Qader, Mir Habibullah Akhgar, Sean Dryer, Mariah Boyd-Boffa, Muluken Melese, Pedro Guillermo Suarez.

**Investigation:** Ghulam Qader Qader, Mir Habibullah Akhgar, Laiqullrahman Rahman.

**Methodology:** Ghulam Qader Qader, Mohammad Khaled Seddiq, Khakerah Mohammad Rashidi, Lutfullah Manzoor, Azizullah Hamim, Mir Habibullah Akhgar, Laiqullrahman Rahman.

**Project administration:** Ghulam Qader Qader, Azizullah Hamim.

**Resources:** Ghulam Qader Qader, Mohammad Khaled Seddiq, Lutfullah Manzoor, Mir Habibullah Akhgar, Laiqullrahman Rahman, Pedro Guillermo Suarez.

**Supervision:** Ghulam Qader Qader, Lutfullah Manzoor, Azizullah Hamim, Mir Habibullah Akhgar, Laiqullrahman Rahman.

**Validation:** Ghulam Qader Qader.

**Visualization:** Ghulam Qader Qader, Pedro Guillermo Suarez.

**Writing – original draft:** Ghulam Qader Qader, Sean Dryer, Mariah Boyd-Boffa, Aleefia Somji, Muluken Melese, Pedro Guillermo Suarez.

**Writing – review & editing:** Ghulam Qader Qader, Sean Dryer, Mariah Boyd-Boffa, Aleefia Somji, Muluken Melese, Pedro Guillermo Suarez.

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
