## [Decision Letter · Decision Letter 0]

4 Feb 2021

PONE-D-20-33514

Prevalence of latent tuberculosis infection among health workers in Afghanistan: a cross-sectional study

PLOS ONE

Dear Author

Thank you for submitting your manuscript to PLOS ONE. After careful consideration, we feel that it has merit but does not fully meet PLOS ONE’s publication criteria as it currently stands. Therefore, we invite you to submit a revised version of the manuscript that addresses the points raised during the review process.

We look forward to receiving your revised manuscript.

Kind regards,

Ramesh Kumar, PhD

Academic Editor

PLOS ONE

Journal Requirements:

2.Thank you for stating the following in the Funding Section of your manuscript:

"The United States Agency for International Development (USAID), through the Challenge TB

project under cooperative agreement number AID-OAA-A-14-00029, and the Global Fund to

Fight AIDS, Tuberculosis and Malaria provided funding for this study."

 "The author(s) received no specific funding for this work"

Additional Editor Comments:

Comments are atteched

Reviewers' comments:

Reviewer's Responses to Questions

**Comments to the Author**

1. Is the manuscript technically sound, and do the data support the conclusions?

Reviewer #1: Yes

Reviewer #2: Yes

2. Has the statistical analysis been performed appropriately and rigorously? 

Reviewer #1: Yes

Reviewer #2: Yes

3. Have the authors made all data underlying the findings in their manuscript fully available?

Reviewer #1: Yes

Reviewer #2: Yes

4. Is the manuscript presented in an intelligible fashion and written in standard English?

Reviewer #1: Yes

Reviewer #2: Yes

5. Review Comments to the Author

Reviewer #1: The authors have selected a very important public health problem i.e. Latent tuberculosis among health workers in Afghanistan. There is need of such studies to generate information for devising national strategy/policy regarding undertaking interventions. Some of the suggested changes before publication are as following:

1. For Reference 1 and 2, please incorporate the access-date and as well as link for the access.

2. In Methods-section, remove the word "we". The sample was calculated on the basis of single proportion formula...please elaborate it. Also explain the systematic random sampling of study facilities and thereafter selection of study participants.

3. In Results-section omit words such as 'majority' in 2nd-paragraph 2nd sentence. Just objectively describe the findings. For tables, briefly explain the findings in caption.

4. In Discussion, also compare the findings with relevant studies in the relevant-settings.

Reviewer #2: This manuscript by Qader etal is a well concieved and well written study that takes a look at prevalence of LTBI in healthcare workers in Afganistan. As pointed out by the authors the main corelation of LTBI with BMI maybe transactional. The main results from the study about prevalence of LTBI are not novel but technically sound as per PLOS one policies.

Minor comment: The authors mention as two places that one third of the world population is suspected LTBI, hoever the they also mention recent WHO data that show that number around one fourh (23%). Authors should reference and use the current data from WHO.

6. PLOS authors have the option to publish the peer review history of their article (what does this mean?). If published, this will include your full peer review and any attached files.

Reviewer #1: No

Reviewer #2: No

---

## [Author Response · Author response to Decision Letter 0]

23 Mar 2021

Reviewer #2

We would like to thank Reviewer #2 for reviewing our article and providing valuable feedback. We have addressed the comments in the following way:

1. Minor comment: The authors mention as two places that one third of the world population is suspected LTBI, however they also mention recent WHO data that show that number around one fourth (23%). Authors should reference and use the current data from WHO.

We updated all the figures with the new WHO report, and actually the current TB infection prediction is 2 billion, which is around 26% of the world population in 2019.

---

## [Decision Letter · Decision Letter 1]

14 May 2021

Prevalence of latent tuberculosis infection among health workers in Afghanistan: a cross-sectional study

PONE-D-20-33514R1

Dear Author,

We’re pleased to inform you that your manuscript has been judged scientifically suitable for publication and will be formally accepted for publication once it meets all outstanding technical requirements.

Kind regards,

Ramesh Kumar, PhD

Academic Editor

PLOS ONE

Additional Editor Comments (optional):

Reviewers' comments:

Reviewer's Responses to Questions

**Comments to the Author**

1. If the authors have adequately addressed your comments raised in a previous round of review and you feel that this manuscript is now acceptable for publication, you may indicate that here to bypass the “Comments to the Author” section, enter your conflict of interest statement in the “Confidential to Editor” section, and submit your "Accept" recommendation.

Reviewer #2: All comments have been addressed

2. Is the manuscript technically sound, and do the data support the conclusions?

Reviewer #2: Yes

3. Has the statistical analysis been performed appropriately and rigorously? 

Reviewer #2: Yes

4. Have the authors made all data underlying the findings in their manuscript fully available?

Reviewer #2: Yes

5. Is the manuscript presented in an intelligible fashion and written in standard English?

Reviewer #2: Yes

6. Review Comments to the Author

Reviewer #2: (No Response)

7. PLOS authors have the option to publish the peer review history of their article (what does this mean?). If published, this will include your full peer review and any attached files.

Reviewer #2: No

---

## [Editor Report · Acceptance letter]

21 May 2021

PONE-D-20-33514R1 

Prevalence of latent tuberculosis infection among health workers in Afghanistan: a cross-sectional study 

Dear Dr. Qader:

I'm pleased to inform you that your manuscript has been deemed suitable for publication in PLOS ONE. Congratulations! Your manuscript is now with our production department. 

Kind regards, 

on behalf of

Dr. Ramesh Kumar 

Academic Editor

PLOS ONE